# Extended Spatially Localized Perturbation GAN (eSLP-GAN) for Robust Adversarial Camouflage Patches [note 1]

**DOI:** 10.3390/s21165323

**Published:** 2021-08-06

**Authors:** Yongsu Kim, Hyoeun Kang, Naufal Suryanto, Harashta Tatimma Larasati, Afifatul Mukaroh, Howon Kim

**Affiliations:** 1SmartM2M, Busan 46300, Korea; yongsu@smartm2m.co.kr; 2Department of Information Convergence Engineering, School of Computer Science and Engineering, Pusan National University, Busan 609735, Korea; hyoeun0915@pusan.ac.kr (H.K.); naufalso@pusan.ac.kr (N.S.); harashta@pusan.ac.kr (H.T.L.); afifatul.mukaroh@pusan.ac.kr (A.M.); 3School of Electrical Engineering and Informatics, Institut Teknologi Bandung, Bandung 40116, Indonesia

**Keywords:** adversarial patch, generative adversarial networks, camouflage

## Abstract

Deep neural networks (DNNs), especially those used in computer vision, are highly vulnerable to adversarial attacks, such as adversarial perturbations and adversarial patches. Adversarial patches, often considered more appropriate for a real-world attack, are attached to the target object or its surroundings to deceive the target system. However, most previous research employed adversarial patches that are conspicuous to human vision, making them easy to identify and counter. Previously, the spatially localized perturbation GAN (SLP-GAN) was proposed, in which the perturbation was only added to the most representative area of the input images, creating a spatially localized adversarial camouflage patch that excels in terms of visual fidelity and is, therefore, difficult to detect by human vision. In this study, the use of the method called eSLP-GAN was extended to deceive classifiers and object detection systems. Specifically, the loss function was modified for greater compatibility with an object-detection model attack and to increase robustness in the real world. Furthermore, the applicability of the proposed method was tested on the CARLA simulator for a more authentic real-world attack scenario.

## 1. Introduction

In addition to their use for computer vision tasks [1], such as facial recognition, object detection [2], and image segmentation, deep neural networks (DNNs) are known for their various potential applications. In particular, the use of DNNs have been found in lane line detection and traffic sign recognition in self-driving cars [3,4,5], Unmanned Aerial Vehicle (UAV)-based monitoring system [6], and even to the usage in the emerging technologies and framework such as big data [7] and blockchain [8] in the industrial network infrastructure. Despite their promising progress, recent research has shown examples of attacks on various intelligent systems [9,10]. In particular, it is known that DNNs included in those systems are vulnerable to adversarial attacks, which lead deep learning models to make incorrect predictions with high confidence.

Adversarial examples are typically created by adding a modicum of noise to the original image. The majority of early research efforts have concentrated on adversarial examples against image classification in digital environments [11,12,13]. Brown et al. [14] proposed a printable universal adversarial patch to deceive a classifier in the physical world. The performance of the classifier was degraded by attaching adversarial patches to a physical object.

However, these adversarial patches could be easily identified by human vision because they prioritized attack performance rather than visual fidelity. Liu et al. [15] proposed a perceptual-sensitive GAN (PS-GAN) for generating adversarial patches through generative adversarial networks (GANs) that could learn and approximate the distribution of original instances. Utilizing a patch-to-patch translation and an attention mechanism, the PS-GAN simultaneously improved the visual fidelity and the capability of the adversarial patch to attack. Despite the considerable effort, adversarial patches generated by a PS-GAN retain an unnatural appearance and, thus, are conspicuous to human vision.

Object detection models have also emerged as critical components of computer vision systems for real-time recognition tasks. Recent studies have revealed that object detection models are vulnerable to adversarial attacks. Eykholt et al. [16] proposed a disappearance attack that caused physical objects to be ignored by a detector in the scene and a creation attack to detect nonexistent objects by using posters or stickers. Zhao et al. [17] implemented adversarial examples that can attack detectors within distances varying from 1 to 25 m and angles from −60° to 60°. However, these object detection model attack methods also have low visual fidelity, similar to the case of the classification model.

Attacks on object detection models are more challenging than those on classification models. While the classifiers focus only on a single label, object detectors would consider multiple labels and the relative positions of the detectors and objects. In addition, because the objects move dynamically, attackers must consider their distance, viewing angle, and illumination.

In our previous work, we proposed a spatially localized perturbation GAN (SLP-GAN) [18] that generated a spatially localized perturbation as an adversarial patch to attack classifiers. It had the advantage that it could generate visually natural patches while maintaining a high attack success rate. The patch region to attach to a target object was extracted from the most representative area of the target object using the Grad-CAM algorithm [19] to improve the ability to attack. However, there are two main limitations of SLP-GAN. First, SLP-GAN can attack only classification models, not object detection models. Second, SLP-GAN has a rather weak attack performance in the physical world because it does not consider the various physical world transformations.

In this paper, we propose an extended spatially localized perturbation GAN (eSLP-GAN) that can attack both classifiers and detectors. We modify the loss function of SLP-GAN to enable attack on the object detection models. We also improve the robustness of adversarial patches by applying terms that consider various transformations and printability in the real-world scenario.

## 2. Related Work

Attacks on deep neural networks (DNNs) have raised concerns for many researchers owing to their potentially fatal impact. Even a slight perturbation added to an image may cause the computer vision model to behave unexpectedly. When successful, this attack type has been proven to disrupt a wide range of computer vision tasks, ranging from simple image classification to object detection models. Recent works have identified the risks posed even in the real-world case, such as adversarial patches being attached to physical objects. In this section, we provide an overview of the state-of-the-art image classification and object detection models that are evaluated using the proposed method. Then, we provide the overview of related adversarial attacks in both the digital and physical domains, as well as generative adversarial networks (GANs), which have recently gained interest for use as methods to generate adversarial attacks.

### 2.1. Classification Model

Image classification or image recognition is the primary task in computer vision. Given image *x*, the goal is to predict the label y∈Y, where *Y* is the set of defined labels corresponding to the image. The convolutional neural network (ConvNet) is the most common neural network architecture used in this field.

VGGNet [20] is a classic Deep ConvNet with its celebrated VGG-16 architecture, that was proposed to learn the effect of convolutional network depth on accuracy. Many researchers have used it as a baseline for large-scale image recognition.

Residual Network (ResNet) architecture [21] introduces a residual or skip connection function to the convolutional neural network that allows the deeper model to be trained. This novel architecture has been an inspiration for later state-of-the-art CNN-based models.

MobileNet [22] and MobileNetv2 [23] are convolution neural networks specifically designed for mobile or embedded vision applications that have limited computing resources. It introduces tunable hyperparameters that efficiently make trade-offs between latency and accuracy.

EfficientNet [24], the state-of-the-art ConvNet, was proposed as a method to scale up the model accuracy effectively by meticulously balancing the network width, depth, and resolutions. The best EfficientNet model can achieve 84.3% top-1 accuracy on a large-scale ImageNet dataset while being 8.4-times smaller and 6.1-times faster than the best-performing ConvNets [24].

### 2.2. Object Detection Model

Object detection is a computer vision task that detects instances of objects of a certain class within an image. Given image *x*, the goal is to predict each object’s location, which is represented as bounding box *B* and label *y* that corresponds to the object. It is also known for combining the object localization and multi-label classification tasks. The common object detection framework consists of a feature extraction module from a pre-trained classification model added to a detection head module. The state-of-the-art object detection approaches can be categorized into two types: two-stage and one-stage object detection.

Two-stage object detection consists of a region of interest (RoI) proposal (where the candidate object is located) and an image classifier to process the candidate region. The R-CNN family [25,26,27,28] represents the most well-known methods using these concepts. The initial R-CNN [25] used a selective search method to propose RoIs and process each candidate region independently, which rendered the model computationally expensive and slow.

Fast R-CNN [26] aggregates the RoI into one unit and uses one CNN forward pass over the entire image, which shares the same extracted feature matrix as the RoI. The shared computation makes the model faster than the original R-CNN; however, it remains inefficient because the RoIs are produced by other models. Faster R-CNN [27] overcomes this issue by unifying the model and proposing a region proposal network (RPN) as the replacement selective-search method.

One-stage object detection is more efficient and has a higher inference speed. It directly predicts the output in a single step. You Only Look Once (YOLO) [29] was an early method for one-stage object detection. The latest version of YOLO, namely the YOLOv4 [30], is proposed as a major improvement to the YOLO family in terms of both speed and accuracy. EfficientDet [31] is another state-of-the-art one-stage object detection method. EfficientDet implements a weighted bi-directional feature pyramid network (BiFPN) for fast multi-scale feature fusion and uses the EfficientNet model as the backbone [31].

### 2.3. Adversarial Attack in Digital Environments

Adversarial attacks in digital environments mainly consist of adversarial perturbation methods that add pixel-level noise to the input images. Adversarial perturbation is tiny perturbation added to the image that causes the target model to misclassify with high confidence [32]. Suppose that a trained model *M* can correctly classify an original image *x* as M(x)=y. By adding to *x* a slight perturbation η, that could not be recognized by human vision, an adversary could generate an adversarial example $x^{\prime }=x+\eta$ such that $M(x^{\prime })\neq y$.

Based on the adversary’s knowledge, adversarial attacks can be categorized as white-box and black-box attacks as follows:White-box attacks: The adversary knows the structure and all the parameters of the target model, including the training data, model architectures, hyper-parameters, and model weights. FGSM [11], Deepfool [33], C&W [12], and PGD [13], are examples of popular techniques for generating adversarial examples based on white-box attacks. Most white-box attack methods rely on model gradients to compute the perturbations.Black-box attacks: The adversary is ignorant of the target model and considers the target as a black-box system. The adversary analyzes the target model by observing only the output based on a given series of adversarial examples. White-box attack methods can be used in black-box attack scenarios by relying on the transferability of adversarial perturbations. The adversary uses a substitute model trained with the same output as the target model and generates adversarial examples with the white-box setting. The generated adversarial example is then used and evaluated in the target model. Using the same approach, we evaluate the transferability of our method to other models. In addition to relying on transferability, some black-box attack methods have been specifically designed for this setting. Gradient-estimation-based [34,35,36] and local-search-based [37,38,39] methods require no knowledge of the target model.

### 2.4. Adversarial Attack in Physical Environments

Adversarial perturbations are typically applied to digital environments in real-world attack scenarios. Conversely, an adversarial patch is feasible in physical environments because it can be attached to a specific location on a real object. Brown et al. [14] introduced a method to create a universal, robust, and targeted adversarial patch for real-world applications. The patch can successfully fool a classifier with a variety of scenes, transformations, and outputs to any target class.

Another proposal is a black-box adversarial-patch-based attack called DPatch. It can fool mainstream object detectors (e.g., Faster R-CNN and YOLO), which cannot be accomplished by the original adversarial patch [40]. It simultaneously attacks the bounding box regression and object classification.

Slightly different from adversarial patches that focus on being applied to physical objects, physical attack methods focus on robustness in the real world. Athalye et al. [41] presented the expectation over transformation (EOT), an algorithm for synthesizing robust adversarial examples over a chosen distribution of transformations. They synthesized adversarial examples that were robust to noise, distortion, and affine transformations and demonstrated the presence of 3D adversarial objects in the physical world.

Eykholt et al. [42] proposed robust physical perturbation ($RP_{2}$), an adversarial attack algorithm in the physical domain that generates robust adversarial perturbations under various physical conditions. The $RP_{2}$ algorithm is an enhanced version of the EOT algorithm that considers printability in the real world. They added a term that models printer color-reproduction errors based on the non-printability score (NPS) by Sharif et al. [43]. They can create adversarial patches that are close to the actual printable colors by minimizing the NPS.

The success of the EOT algorithm was also proven by Chen et al. [44], who proposed a robust physical adversarial attack on a Faster R-CNN object detector called ShapeShifter. The attack shows that EOT can also be applied in object detection tasks and significantly enhances the robustness of the resulting perturbation captured under different conditions. They tested the attack on real-world driving and showed that the perturbed stop signs can constantly fool the Faster R-CNN object detector. However, most adversarial patches or physical perturbation methods produce a visible pattern that can be easily recognized by human vision. Therefore, we focus on the invisibility of the patch while maintaining robustness in a real-world implementation by utilizing the EOT and $RP_{2}$ algorithms for our eSLP-GAN.

### 2.5. Generative Adversarial Networks (GANs)

Recent studies have shown that adversarial perturbation and patch generation mostly rely on optimization schemes. To generate a perceptually more realistic perturbation efficiently, researchers have proposed many variants of generative adversarial networks (GANs) [45]. Earlier, Goodfellow et al. [45] introduced a framework for estimating generative models via an adversarial process, which simultaneously trained two models: a generative model *G* that captures the data distribution, and a discriminative model *D* that estimates the probability that a sample came from the training data rather than *G*. This discovery, coined as a generative adversarial network (GAN), has had a significant impact on data generation.

In the context of adversarial perturbation generation, Xiao et al. [46] proposed Adv-GAN to generate adversarial perturbations using generative adversarial networks (GANs). It can learn and approximate the distribution of the original instances. Once the generator is trained, it can generate perturbations efficiently for any instance to potentially accelerate adversarial training for defense. This attack was placed first with an accuracy of 92.76% in a public MNIST black-box attack challenge. Liu et al. [15] proposed a perceptual-sensitive GAN (PS-GAN) that simultaneously enhances the visual fidelity and the attacking ability of an adversarial patch.

To improve the visual fidelity, we treat patch generation as a patch-to-patch translation via an adversarial process, feeding a seed patch, and outputting a similar adversarial patch that has a moderately high perceptual correlation with the attacked image. To further enhance the attacking ability, an attention mechanism coupled with adversarial generation was introduced to predict the critical attacking areas for patch placement. The limitation of PS-GAN is that it uses a seed patch that is quite different from the original image and, thus, may still not seem visually natural. The proposed method generates a patch from the original input image using an attention map that maximizes the attack rate while maintaining high visual fidelity.

## 3. Proposed Method

In this section, we describe the problem definition for attacking both the classification model and the object detection model and introduce the extended spatially localized perturbation GAN (eSLP-GAN) framework for generating robust adversarial camouflage patches. All mathematical notations used in eSLP-GAN can be referred to in Table 1.

### 3.1. Problem Definition

This section defines the problem of generating adversarial camouflage patches for classification and object detection models. Assume that $h_{\theta }$ denotes a hypothesis function capable of performing a classification or object detection task. *x* denotes input data to $h_{\theta }$ with a corresponding label of *y*, which satisfies the equation $h_{\theta }\left(x\right)=y$. The label is constructed differently depending on the task. There are two types of attack methods: untargeted and targeted. The purpose of the adversarial attack in the proposed approach is to generate the adversarial example $x_{A}$. The untargeted attack can be expressed as $h_{\theta }\left(x_{A}\right)\neq y$.

The targeted attack is described as $h_{\theta }\left(x_{A}\right)=y_{t}$, where $y_{t}$ is the attack target label. In addition, $x_{A}$ should be comparable to the original input data *x* in terms of visual fidelity. The adversarial example $x_{A}$ is constructed by attaching a spatially localized patch *p* to the original input data *x*, as specified by the equation $x_{A}=x+p$. In the study’s terminology, “spatially localized” refers to the property of perturbation that is applied to a subset of the input image rather than the entire image. The following section describes the proposed method for generating adversarial camouflage patches that meet the aforementioned requirements.

### 3.2. eSLP-GAN Framework

#### 3.2.1. eSLP-GAN

The proposed eSLP-GAN architecture consists of three primary components: a generator *G*, a discriminator *D*, and a target model *M*. The target model *M* approximates the hypothesis function $h_{\theta }$ that performs the classification task or the object detection task mentioned above. Generator *G* takes the original data *x* as input and generates a perturbation over the entire region of the input image. In contrast to previous proposals [46], this method employs additional steps to generate spatially localized perturbations for adversarial camouflage patches that can be attached to a specific location.

We leverage the Grad-CAM algorithm [19] to extract patch regions and apply the generated perturbation to only the extracted region. The method is used for extracting the most representative area that affects the target model to determine the input image as a specific label. Our assumption is that attack performance will be improved in the case of attaching the adversarial patch to the representative area from an input image rather than attaching it to another area. As a result, spatially localized perturbations can be treated as adversarial patches *p*.

The discriminator *D* is responsible for differentiating the original data *x* and the adversarial example $x_{A}=x+p$. *D* encourages *G* to generate perturbation that visually conforms to the original data. Furthermore, *G* should be capable of deceiving the target model *M*. Thus, the entire structure of eSLP-GAN has four loss functions: adversarial loss $L_{ADV}$, attacking ability loss $L_{ATK}$, perturbation loss $L_{PTB}$, and printability loss $L_{NPS}$. The adversarial loss $L_{ADV}$ is expressed by the following equation:(1)$$L_{ADV}=E_{x}\log D\left(x\right)+E_{x}\log\left(1-D\left(x_{A}\right)\right).$$

As observed in the preceding equation, discriminator *D* aims to distinguish the adversarial example $x_{A}$ from the original data *x*. Note that *D* promotes *G* to generate perturbation with visual fidelity in compliance with the above equation.

The attacking loss $L_{ATK}$ is configured differently depending on the attack method and the type of target model *M*. We used the EOT algorithm to construct the attacking loss to improve robustness in real-world situations. In the case where *M* is a classification model, the attacking loss can be defined as follows:(2)$$L_{ATK}=\left\{\begin{array}{cc} -E_{x,t∼T}\ell_{cls}\left(M\left(t\left(x_{A}\right)\right),y\right), & \mathrm{if}\;\mathrm{untargeted}\;\mathrm{attack}.\\ E_{x,t∼T}\ell_{cls}\left(M\left(t\left(x_{A}\right)\right),y_{t}\right), & \mathrm{otherwise}. \end{array}\right.$$
where *y* is the original label of the input data *x* and $y_{t}$ is the target label. $\ell_{cls}$ is the classification loss function (e.g., cross-entropy loss) applied to the target model *M*. We use a chosen distribution *T* of transformation functions *t*, and fool the target model to misclassify $M(x_{A})$ or classify $M(x_{A})$ as the target label $y_{t}$ by minimizing $L_{ATK}$. Differently expressed, we can generate robust adversarial examples that remain adversarial under image transformations that occur in the real world, such as angle and viewpoint changes.

In the case of the object detection model, it predicts bounding boxes that determine the location of objects in an input image. In general, each bounding box contains the objectness score, box position, and class probability vector [47]. We use two types of object detection losses: $\ell_{obj}$ and $\ell_{cls}$. $\ell_{obj}$ represents the maximum objectness score over the predictions for the entire image. $\ell_{cls}$ is the classification loss, which is the concept also used in the classification model. We define the attacking loss $L_{ATK}$ using $\ell_{obj}$ and $\ell_{cls}$ in the case where *M* is an object detection model as follows:(3)$$L_{ATK}=\left\{\begin{array}{cc} \ell_{obj}-E_{x,t∼T}\ell_{cls}\left(M\left(t\left(x_{A}\right)\right),y\right), & \mathrm{if}\;\mathrm{untargeted}\;\mathrm{attack}.\\ -\ell_{obj}+E_{x,t∼T}\ell_{cls}\left(M\left(t\left(x_{A}\right)\right),y_{t}\right), & \mathrm{otherwise}. \end{array}\right.$$

This differs from the attacking loss of the classification model by the addition of the objectness score. In the case of an untargeted attack, the target model considers the bounding box empty of objects by minimizing $\ell_{obj}$. In contrast, in the case of a targeted attack, the target model is induced to classify $x_{A}$ as the target label $y_{t}$ by minimizing $\ell_{cls}$, and the objectness score for the target label is increased by maximizing $\ell_{obj}$.

Additionally, we define the perturbation loss $L_{PTB}$ using a soft hinge loss [48] on the $L_{2}$ norm to bound the magnitude of the generated perturbation as follows:(4)$$L_{PTB}=E_{x}\max\left(0,{∥G(x)∥}_{2}-c\right),$$
where *c* represents the user-specified bound and serves to stabilize the GAN training. To account for printability, we add a term $L_{NPS}$ to our objective function that models the fabrication error. This term is based on the non-printability score (NPS) by Sharif et al. [43].
(5)$$L_{NPS}=\prod_{p^{\prime }\in P^{\prime }}\left|p_{i}-p^{\prime }\right|,$$
where $p^{\prime }\in P^{\prime }$ represents the printable RGB triplets and $p_{i}$ is each pixel of the adversarial patch *p*. We can generate adversarial patches close to printable color by minimizing $L_{NPS}$. Finally, combined with the above visual fidelity and ability to attack the target model, the eSLP-GAN loss function can be expressed as follows:(6)$$L=L_{ADV}+\alpha L_{ATK}+\beta L_{PTB}+\gamma L_{NPS},$$
where α>0, β>0, and γ>0 control the contribution of each loss function. In our eSLP-GAN, the generator *G* and discriminator *D* are trained by solving a minimax game denoted by the equation $\min_{G}\max_{D}L$. As a result, generator *G* can generate spatially localized perturbations suitable for adversarial patches while maintaining visual fidelity and attacking ability. The overall architecture of the eSLP-GAN is depicted in Figure 1, and Algorithm 1 describes the process for training the eSLP-GAN framework.
**Algorithm 1** Training process of the eSLP-GAN Framework. **Input:** training image set $X_{image}=\left\{x_{i}|i=1,\ldots ,n\right\}$ **Output:** spatially localized patches $P=\{p_{i}|i=1,\ldots ,n\}$ **for** the number of training epochs **do**  **for**
*k* steps **do**   sample minibatch of *m* images $\phi_{x}=\left\{x_{1},\ldots ,x_{m}\right\}$.   generate minibatch of *m* adversarial perturbations $\phi_{x}^{G}=\left\{G\left(x_{1}\right),\ldots ,G\left(x_{m}\right)\right\}$.   obtain activation maps $M(\phi_{x})$ by *Grad-CAM*.   extract spatially localized patches $P=\{G{\left(x_{i}\right)}_{M(x_{i})}|i=1,\ldots ,n\}$.   create adversarial examples $x_{A}=\left\{x_{i}+p_{i}|i=1,\ldots ,n\right\}$.   update *D* to $\max_{D}L$ with *G* fixed.  **end for**  sample minibatch of *m* images $\phi_{x}=\left\{x_{1},\ldots ,x_{m}\right\}$.  create adversarial examples $x_{A}$ (same as above).  update *G* to $\min_{G}L$ with *D* fixed. **end for**

#### 3.2.2. Spatial Localization

Grad-CAM [19] is suitable for extracting a representative area from an input image. Grad-CAM produces “visual explanations” from convolutional neural network (CNN)-based models. It generates a localization map highlighting the important regions in the input image by using the gradients of any target label flowing into the final convolutional layer. To obtain a localization map according to the target label (class), defined as a class-discriminative localization map, we used a gradient of the score for class $y^{c}$ with respect to the feature map activation $A^{k}$ of the target layer.
(7)$$\alpha_{k}^{c}=\frac{1}{Z}\sum_{i}\sum_{j}\frac{\partial y^{c}}{\partial A_{ij}^{k}}$$

Here, $\alpha_{k}^{c}$ indicates the neuron importance weight associated with feature map *k* for target class *c*. We take a weighted sum of the forward activation maps, A, with weights $\alpha_{k}^{c}$, and combine them with a ReLU to obtain counterfactual explanations that have a positive effect on the target class.
(8)$$L_{Grad-CAM}^{c}=ReLU\left(\sum_{k}\alpha_{k}^{c}A^{k}\right)$$

After obtaining a localization map with class activation mapping (CAM), we take the bounding boxes for the activated regions by filtering for values greater than 80% of the CAM intensity. The acquired bounding boxes are the most representative areas of the target model decision for the input image. Figure 2 shows examples of Grad-CAM visualizations with bounding boxes for traffic signs.

## 4. Experiments

### 4.1. Experimental Setup

#### 4.1.1. Model Structure

To configure our eSLP-GAN framework, we utilize structures for generator *G* and discriminator *D* with image-to-image translation, as in [48,49]. Generator *G* consists of a U-Net [50] structure, which has an encoder–decoder network. U-Net has the advantage of generating high-resolution images by adding skip connections to construct a contracting path to capture the context and a symmetric expanding path that enables precise localization.

We adopted U-Net architecture as a generator because it is suitable for generating adversarial patches not only similar to input image but also with a relatively good attack performance. This is due to its ability to capture the context, enabling it to extract information from the low level to the high level of the input image. Furthermore, discriminator *D* uses a common CNN architecture in the form of Convolution–BatchNorm–ReLU to encourage generator *G* to increase the generation ability.

#### 4.1.2. Target Models

In this experiment, we used two types of target models: a classification model and an object detection model. As the classification models, we used VGG16 [20], ResNet50 [21], MobileNetV2 [23], and EfficientNetB0 [24], which exhibit good performance in image classification problems. For object detection, we used Faster R-CNN [27], YOLOv4 [30], and EfficientDetD0 [31], which are state-of-the-art object detection models.

#### 4.1.3. Implementation Details

For the implementation, we utilized PyTorch, with testing performed on a Linux workstation (Ubuntu 18.04) with four NVIDIA Titan XP GPUs, Intel Core i9-7980XE CPU, and 64 GB of RAM. We trained the eSLP-GAN for 250 epochs with a batch size of 128, with an initial learning rate of 0.001, and dropped it by 10% every 50 epochs.

### 4.2. Experiment Results

#### 4.2.1. Classification Models

In the case of classification models, we conducted a physical-world attack experiment to verify that the eSLP-GAN was enhanced to be more robust to the physical environment than the previous version. We used Korean traffic sign mock ups approximately 15 × 15 cm in size and from nine classes {Bike Path, No Entry, Danger, Motorway, No Bicycle, No Jaywalking, Roadworks, Railroad Crossing, and School Zone}, as shown in Table 2.

We first took approximately 150 pictures of each traffic sign mock up at varying distances {1, 1.25, 1.5, 1.75, and 2 m} and angles {0°, 15°, 30°, −15°, and −30°}. Figure 3 shows each point at which traffic sign images with different distances and angles were taken. We verified the robustness of the eSLP-GAN under various physical conditions by altering the environmental settings with different distances and angles.

Next, we trained the VGG16, ResNet50, MobileNetV2, and EfficientNetB0 as target models using these images that were divided into a training dataset (approximately 100 images) and a validation dataset (approximately 50 images). To configure a test dataset for both the original images and the adversarial examples equivalently, we randomly selected again approximately 100 pictures as the test dataset, and adversarial patches were generated using eSLP-GAN for each image and each target model.

In this case, we generated half of the patches using an untargeted attack and the remainder using a targeted attack by selecting a random label. After printing these patches, we attached them to the traffic sign mock ups and again took photos of each mock up with the patches attached. Table 3 presents the classification accuracy of the target models on normal validation images and the adversarial examples with attached patches using the test dataset.

Table 4 shows examples of traffic sign mock ups with and without adversarial patches and the classification results. The previous SLP-GAN version had an attack performance that reduced the classification accuracy of the target model from 97.8% to 38.0% [18]. We observe in Table 3 that the enhanced version, eSLP-GAN, had a higher attack performance than the previous version. In other words, eSLP-GAN is robust against real-world distortions, such as lighting conditions, various distances, and angles, by additionally applying the EOT algorithm and NPS loss.

Table 5 represents the physical-world attack result examples of the Roadworks’ traffic sign mock ups with various transformations on the VGG16 target model. Table 6 presents the attack results of the traffic sign mock ups of a targeted attack and untargeted attack at different distances and angles. We can conduct both targeted and untargeted attacks by adjusting the attacking loss term of the eSLP-GAN, as shown in Equation (Equation 2). In the case of Table 6, we set the target label as “Motorway”.

We also conducted transferability experiments between each target model. Table 7 shows the classification accuracy of each target model using transferability. First, we generate adversarial patches for the source models and, then, attached these patches to traffic sign mock ups. Note that adversarial examples against other source models significantly reduced the classification accuracy of the target models. This means that our eSLP-GAN encourages transferability among different target models in physical environments.

#### 4.2.2. Object Detection Models

For the classification models, we evaluated the attack performance of eSLP-GAN in a real-world environment using custom-trained models with traffic sign mock ups. In the case of the object detection models, we evaluated eSLP-GAN by expanding the scope of pre-trained models with large datasets, such as COCO [51]. In an attack scenario of object detection models, we applied an untargeted attack to classify a traffic sign as a different class or a targeted attack to classify a specific class for a label existing in the COCO dataset. Instead of various restrictions, such as financial and time constraints existing in the real world, we used a CARLA simulator [52] that was photo-realistic to increase the efficiency of the experiment. The CARLA simulator is built on the Unreal Engine and provides various maps, vehicle models, and traffic signs, in addition to an autonomous driving system.

We experimented using various maps and traffic signs from the CARLA simulator, as shown in Figure 4. First, we captured image datasets of various traffic signs using the CARLA simulator. Next, we added the “traffic sign” label to the object detection models pre-trained on the COCO dataset and fine-tuned the models using the collected traffic signs image dataset. Next, we generated adversarial camouflage patches using eSLP-GAN extracted for the traffic sign labels. Here, to reproduce the experiment in the real world, the patches were not applied directly to the image, but we used a method of attaching the patch to the corresponding area in the texture map of the traffic sign object. That is, we use an approach similar to attaching a patch directly to a traffic sign in the real world.

We collected approximately 1000 images of traffic signs at various locations in the CARLA simulator, and we split the dataset into 700 training images, 200 validation images, and 100 test images. Next, we used the training datasets and validation datasets for fine-tuning the object detection models, and we used the test dataset to evaluate the object detection models as shown in Table 8 and Table 9. To illustrate an object detection model attack used in autonomous driving, we captured images of traffic signs from the perspective of a vehicle running in a lane.

Next, we measured the mean average precision (mAP) with the IoU threshold set to 0.5 of the object detection model for traffic signs by attaching patches that were generated by the eSLP-GAN to target each object detection model. We used state-of-the-art object detection models, Faster R-CNN [27], YOLOv4 [30], and EfficientDet [31]. Figure 5 illustrates the collected original images and adversarial examples with patches, while Table 8 details the mAP of each object detection model. As shown, our eSLP-GAN performed well against both the classification and object detection models.

As with the classification models, we experimented with the transferability in object detection models. We collected approximately 100 adversarial examples to be used against each source model and evaluated the mAP of each target model. Table 9 lists the mAP values of each target model. Our eSLP-GAN also encouraged transferability in object detection models.

## 5. Conclusions

In this paper, we proposed an extended spatially localized perturbation GAN (eSLP-GAN) for generating adversarial camouflage patches. We improved the robustness of the real-world attack over the previous version and extended it to attack an object detection model. We extracted the target region to place adversarial patches that could maximize the attack success rate by using the Grad-CAM algorithm. In addition, we generated robust adversarial patches in the real world by combining the EOT algorithm and NPS loss and attacked the object detection models through an appropriate loss function correction for the eSLP-GAN. The experimental results show that eSLP-GAN generated robust adversarial patches that were visually natural with a high attack performance that can be used for both classification and object detection models.

The proposed eSLP-GAN was validated on the custom traffic sign dataset against the given classification models and showed that the attack performance improved compared to the previous version. The proposed method also performed well in a black-box attack because, as can be observed from the experimental results, it encouraged transferability. We also verified the efficacy of attacking object detection models using a photo-realistic simulator.

The proposed methods emphasize the vulnerability of computer vision models that are widely used in the real world. These attacks, which are difficult to detect with human vision, are especially fatal to computer vision systems because they are concealed. Therefore, it is suggested that improvements to the security of computer vision models are essential to withstand attacks, such as by the proposed method. 

## Figures and Tables

**Figure 1 sensors-21-05323-f001:**
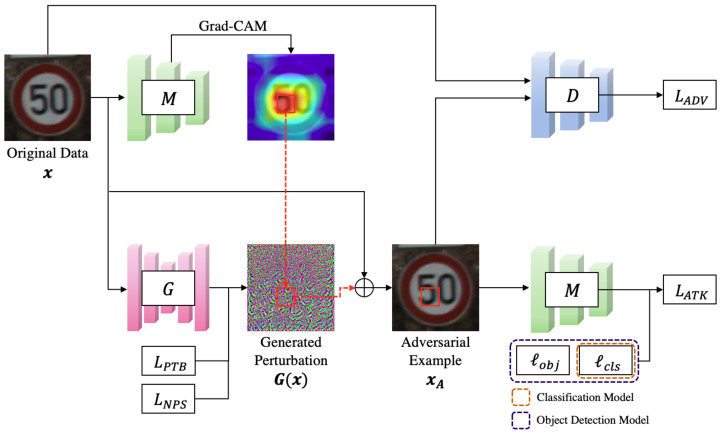
The eSLP-GAN framework consists of the generator *G*, the discriminator *D*, and the target model *M*.

**Figure 2 sensors-21-05323-f002:**
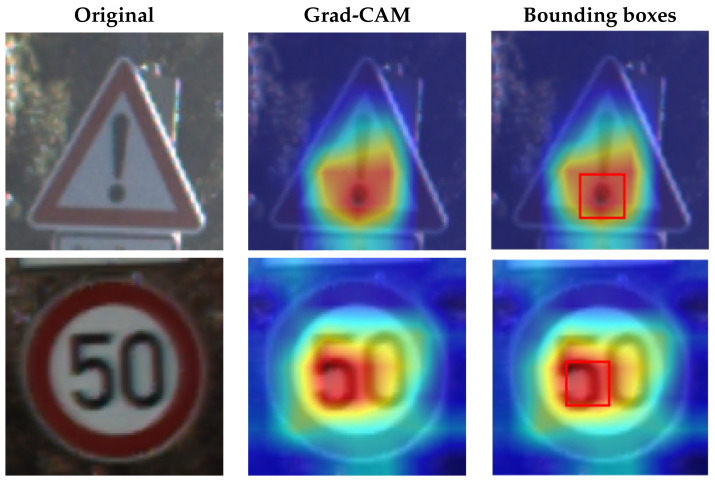
Samples of Grad-CAM visualizations and bounding boxes on traffic signs.

**Figure 3 sensors-21-05323-f003:**
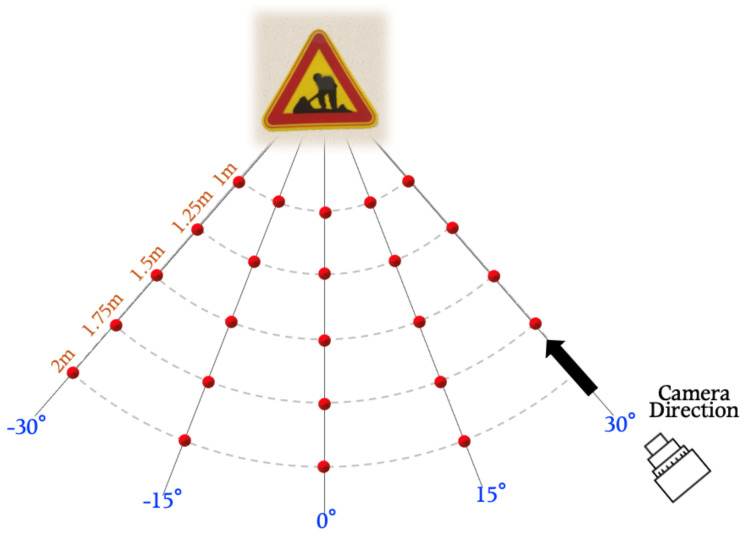
Physical-world attack experiment settings against classification models with different distances and angles.

**Figure 4 sensors-21-05323-f004:**
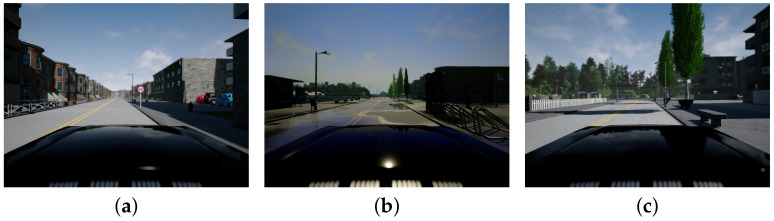
Examples of photo-realistic environments in the CARLA simulator. (**a**–**c**) show the capture of some different panoramas and shades in the CARLA simulator.

**Figure 5 sensors-21-05323-f005:**
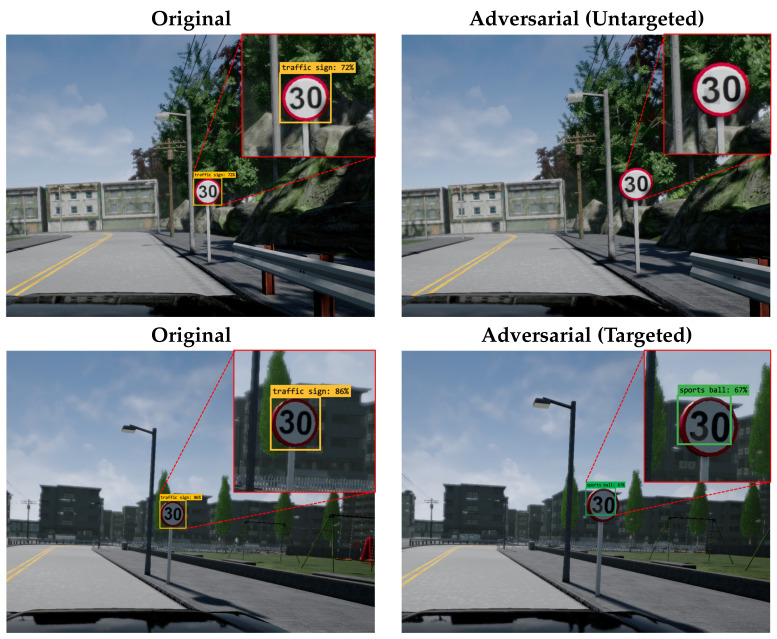
Samples of original images and adversarial examples for traffic signs.

**Table 1 sensors-21-05323-t001:** List of symbols in this paper.

Symbol	Meaning	Symbol	Meaning
$h_{\theta }$	Hypothesis function of classification or object detection task	*L*	Total loss function of eSLP-GAN
*x*	Input data	$L_{ADV}$	Adversarial loss function of eSLP-GAN
$x_{A}$	Adversarial example	$L_{ATK}$	Attacking ability loss function of eSLP-GAN
*y*	Ground truth (label) of input data *x*	$L_{PTB}$	Perturbation loss function of eSLP-GAN
$y_{t}$	Attack target label	$L_{NPS}$	Printability loss function of eSLP-GAN
*p*	Spatially localized patch	$l_{cls}$	Classification loss function applied to the target model *M*
*G*	Generator of eSLP-GAN	$l_{obj}$	Objectness loss function applied to the target model *M*
*D*	Discriminator of eSLP-GAN	*T*	Chosen distribution of transformations
*M*	Attack target model	*t*	Transformation functions

**Table 2 sensors-21-05323-t002:** Korean traffic sign mock ups used in the physical-world attack experiment against classification models.

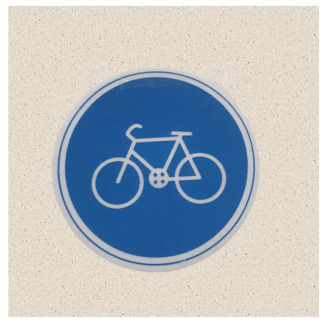	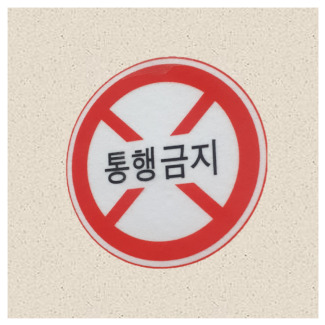	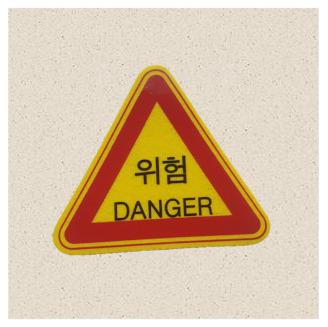
Bike Path	No Entry	Danger
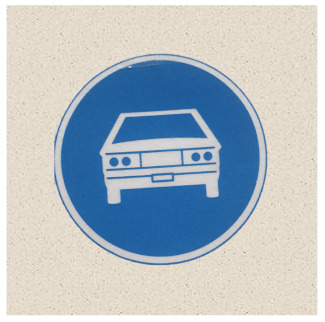	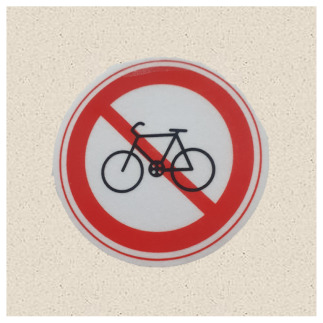	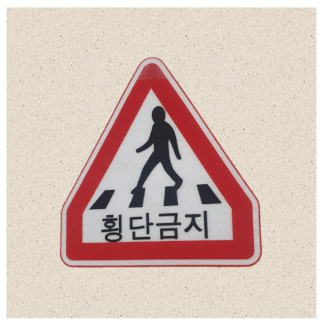
Motorway	No Bicycle	No Jaywalking
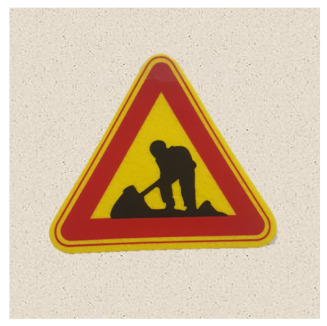	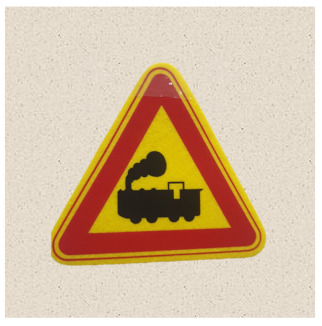	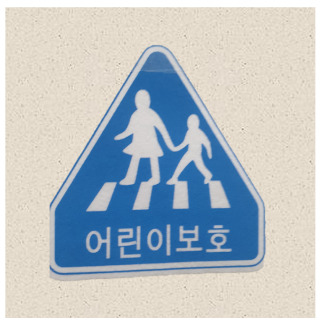
Roadworks	Railroad Crossing	School Zone

**Table 3 sensors-21-05323-t003:** Classification accuracy of target classification models on normal images and adversarial examples.

Target Model	Classification Accuracy	
	Original	Adversarial
VGG16	98.24%	**31.49%**
ResNet50	97.48%	**29.81%**
MobileNetV2	96.91%	**36.58%**
EfficientNetB0	96.35%	**34.24%**

**Table 4 sensors-21-05323-t004:** The physical-world attack result examples of traffic sign mock ups with and without adversarial patches generated by eSLP-GAN.

Original	Adversarial	Original	Adversarial
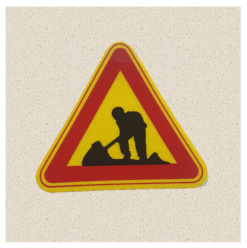	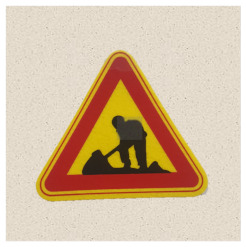	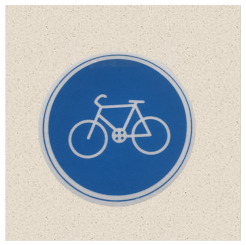	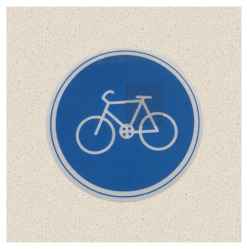
Roadworks	Danger	Bike Path	No Jaywalking
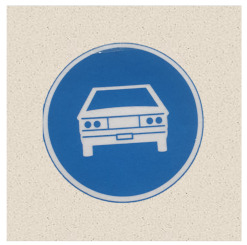	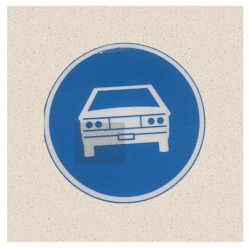	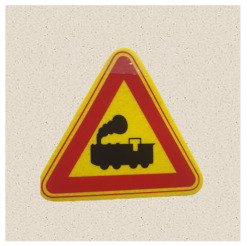	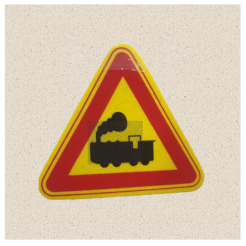
Motorway	School Zone	Railroad Crossing	No Entry

**Table 5 sensors-21-05323-t005:** The physical-world attack result examples of ‘Roadworks’ traffic sign mock ups with various transformations.

Original	Adversarial	
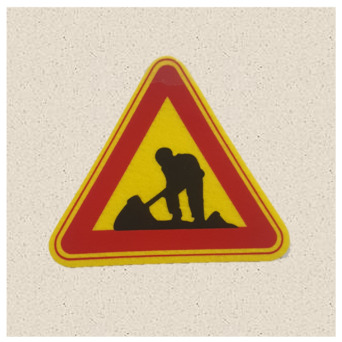	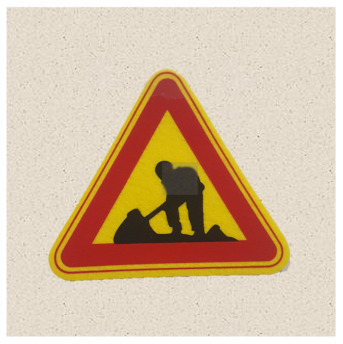	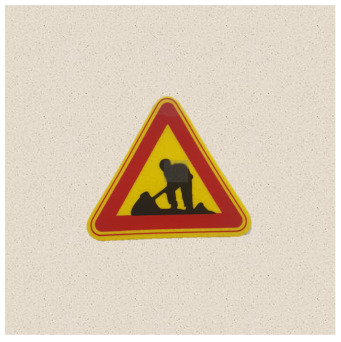
Roadworks	Danger	Danger
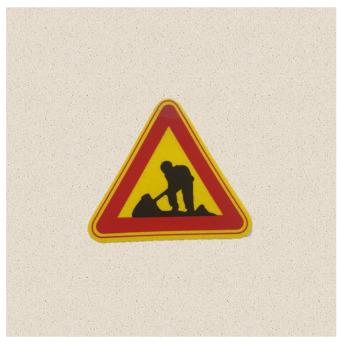	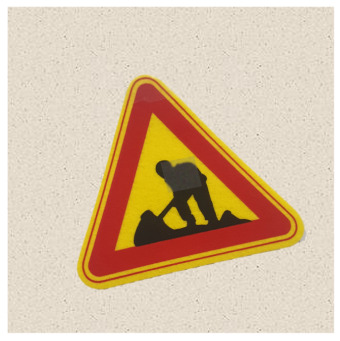	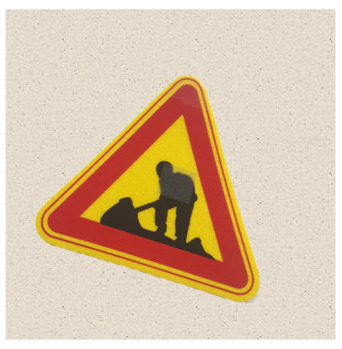
Roadworks	Danger	Roadworks

**Table 6 sensors-21-05323-t006:** The physical-world attack result of ‘Roadworks’ traffic sign mock ups of targeted attack and untargeted attacks with different distances and angles against VGG16.

Distance	Angle	Dangers	(Conf.)	Motorway	(Conf.)	Untargeted	(Conf.)
1 m	30°	Dangers	(0.71)	Motorway	(0.85)	Roadworks	(0.88)
	15°	Dangers	(0.93)	Motorway	(0.87)	Railroad Crossing	(0.78)
	0°	Dangers	(0.87)	Motorway	(0.93)	Railroad Crossing	(0.64)
	−15°	Dangers	(0.90)	Motorway	(0.61)	Railroad Crossing	(0.73)
	−30°	Roadworks	(0.91)	Motorway	(0.70)	Roadworks	(0.87)
1.25 m	30°	Dangers	(0.81)	Motorway	(0.65)	Danger	(0.91)
	15°	Dangers	(0.84)	Motorway	(0.70)	Danger	(0.83)
	0°	Dangers	(0.87)	Roadworks	(0.75)	Roadworks	(0.58)
	−15°	Dangers	(0.86)	Motorway	(0.71)	Danger	(0.64)
	−30°	Dangers	(0.77)	No Entry	(0.40)	Railroad Crossing	(0.70)
1.5 m	30°	Dangers	(0.78)	Motorway	(0.85)	Railroad Crossing	(0.91)
	15°	Dangers	(0.79)	Motorway	(0.91)	Railroad Crossing	(0.91)
	0°	Dangers	(0.80)	Motorway	(0.89)	Roadworks	(0.62)
	−15°	Dangers	(0.74)	Roadworks	(0.68)	Bike path	(0.80)
	−30°	Dangers	(0.68)	Motorway	(0.89)	No Jaywalking	(0.70)
1.75 m	30°	Dangers	(0.61)	Motorway	(0.68)	No Jaywalking	(0.63)
	15°	Dangers	(0.58)	Motorway	(0.62)	Dangers	(0.81)
	0°	Roadworks	(0.55)	Motorway	(0.61)	Railroad Crossing	(0.93)
	−15°	Dangers	(0.56)	Motorway	(0.61)	Railroad Crossing	(0.96)
	−30°	Dangers	(0.61)	No Entry	(0.55)	No Entry	(0.91)
2 m	30°	Motorway	(0.69)	Motorway	(0.60)	Railroad Crossing	(0.96)
	15°	Dangers	(0.63)	Motorway	(0.61)	Railroad Crossing	(0.94)
	0°	Roadworks	(0.51)	Motorway	(0.55)	Roadworks	(0.94)
	−15°	Dangers	(0.61)	Motorway	(0.58)	Danger	(0.91)
	−30°	Railroad Crossing	(0.55)	Motorway	(0.51)	Railroad Crossing	(0.98)
Success rates			73%		67%		78%

**Table 7 sensors-21-05323-t007:** Classification accuracy of each target classification model using the transferability.

		Target Models			
		VGG	ResNet	MobileNet	EfficientNet
**Source Models**	VGG	**31.49%**	57.32%	49.81%	54.38%
	ResNet	42.78%	**29.81%**	60.57%	53.13%
	MobileNet	56.04%	47.28%	**36.58%**	55.91%
	EfficientNet	44.38%	53.12%	50.45%	**34.24%**

**Table 8 sensors-21-05323-t008:** mAP of target object detection models on normal images and adversarial examples.

Target Model	mAP	
	Original	Adversarial
Faster R-CNN	83.74%	**48.91%**
EfficientDet-D0	97.27%	**55.38%**
YOLOv4	98.13%	**61.55%**

**Table 9 sensors-21-05323-t009:** mAP of each target object detection model using the transferability.

		Target Models		
		Faster R-CNN	EfficientDet-D0	YOLOv4
**Source** **Models**	Faster R-CNN	**48.91%**	73.58%	75.39%
	EfficientDet-D0	62.14%	**55.38%**	72.15%
	YOLOv4	60.48%	71.38%	**61.55%**

## Data Availability

Not applicable.

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
