# Peer review of "Extended Spatially Localized Perturbation GAN (eSLP-GAN) for Robust Adversarial Camouflage Patches†"

_sensors, 2021, doi:10.3390/s21165323_

Round 1

Reviewer 1 Report

In this paper, the use of the method called eSLP-GAN was extended to deceive classififiers and object detection systems. Specifically, the loss function was modified for greater compatibility with an object-detection model attack and to increase robustness in the real world. Furthermore, the applicability of the proposed method was tested on the CARLA simulator for a more authentic real-world attack scenario. This paper discusses an interesting problem, and the structure is good. However, a mojor revision is needed before the acceptance.

1) This work looks like is based on another research work called SLP-GAN. The details and differences about SLP-GAN should be presented to the the research background clearer.

2) A lot of well-studied technologies (e.g., Grad-CAM, U-Net) are used in the proposed framework, and what is the main key contribution of this work. It needs to be emphasized.

3) For the evaluation, describe more details of the tested datasets, and the machine/platform configuration can be introduced.

4) Remove all the typographic errors (e.g., “which leads deep learning models to make” -> “which lead deep learning models to make”). Carefully check throughout the paper.

5) A symbol table is needed to explain all used mathematical notations.

6) The Reference is sort of messy, some items (e.g., [1], [4]) lack the necessary information. Please double check this part, and follow the right template.

7) Make the References more comprehensive, besides the adversarial attacks, the similar work may can be applied in some other promising scenarios (e.g., Blockchain, Big Data or other IoT systems). If the above related work can be discussed, it can strongly improve the research significance. For the improvement, the following papers can be considered to make the references more comprehensive.

  • A. Qayyum, I. Ahmad, M. Iftikhar and M. Mazher, “Object detection and fuzzy-based classification using uav data,” Intelligent Automation & Soft Computing, vol. 26, no.4, pp. 693–702, 2020.
  • F. Zhang, H. Zhao, W. Ying, Q. Liu, A. Noel, et al., "Human face sketch to rgb image with edge optimization and generative adversarial networks," Intelligent Automation & Soft Computing, vol. 26, no.6, pp. 1391–1401, 2020.
  • M. B. Nejad B and M. E. Shir, “A new enhanced learning approach to automatic image classification based on SALP swarm algorithm,” Computer Systems Science and Engineering, vol. 34, no.2, pp. 91–100, 2019.
  • F. N. Al-Wesabi, H. G. Iskandar, M. Alamgeer and M. M. Ghilan, "Proposing a high-robust approach for detecting the tampering attacks on english text transmitted via internet," Intelligent Automation & Soft Computing, vol. 26, no.6, pp. 1267–1283, 2020.
  • J. Wang, Y. Yang, T. Wang, R. Sherratt, J. Zhang. Big Data Service Architecture: A Survey. Journal of Internet Technology, 2020, 21(2): 393-405
  • Z. Gu, Y. Su, C. Liu, Y. Lyu, Y. Jian et al., “Adversarial attacks on license plate recognition systems,” Computers, Materials & Continua, vol. 65, no. 2, pp. 1437–1452, 2020.
  • J. Zhang, S. Zhong, T. Wang, H.-C. Chao, J. Wang. Blockchain-Based Systems and Applications: A Survey. Journal of Internet Technology, 2020, 21(1): 1-14
  • Y. Lee, H. Ahn, H. Ahn and S. Lee, “Visual object detection and tracking using analytical learning approach of validity level,” Intelligent Automation & Soft Computing, vol. 25, no.1, pp. 205–215, 2019.

Reviewer 2 Report

In this paper the authors propose and test new spatially localized perturbations GAN as adversarial patches against computer vision, object detection and classification models. The paper reads well, has nice experimentation and contribution. In my opinion it should be published after a few modifications, i.e.:

  • In the abstract, the authors say that DNN are primarily used in computer vision. This is not valid and the authors need to revise or remove this claim.
  • In section 4.1.3 the authors should clarify why the experiment parameters. For example, why the learning rate was dropped by 10% (and not 12%) every 50 epochs (again why not 55 epochs).
  • In Section 2.3, the authors should provide more details on transferability experiments that they have conducted.

Round 2

Reviewer 1 Report

Revised paper is good enough and can be accepted now.

This manuscript is a resubmission of an earlier submission. The following is a list of the peer review reports and author responses from that submission.